# Temporal Energy Transformer for Long Range Propagation in Continuous Time Dynamic Graphs

**Parveena Shamim Abdul Salam**                     *parveenashamim.a@fujitsu.com*
*Fujitsu Research of India*
**Abhishek Ajayakumar**                              *abhishek.a@fujitsu.com*
*Fujitsu Research of India*
**Harsh Pandey**                                     *harsh.p@fujitsu.com*
*Fujitsu Research of India*

**Reviewed on OpenReview:** *https://openreview.net/forum?id=zg3bi0GRJk*

## Abstract

Representation learning on temporal graphs is crucial for understanding dynamically varying real-world systems such as social media platforms, financial transactions, transportation networks, and communication systems. Existing self-attention based models encounter limitations in capturing long-range dependencies and lack clear theoretical foundations. Energy-based models offer a promising alternative, with a well-established theoretical foundation that avoids reliance on pseudo-losses. However, their application in this domain remains largely unexplored, primarily due to the challenge of designing energy functionals. In this work, we introduce the Temporal Energy Transformer (TET), a novel energy-based architecture that integrates with the Temporal Graph Network (TGN) framework. Our approach centres on a novel energy-based graph propagation module that leverages a specially designed energy functional to capture and preserve spatio-temporal information. This is achieved by modelling the temporal dynamics of irregular data streams with a continuous-time differential equation. Our temporal energy transformer (TET) layer employs a series of temporal energy attention layers and a dense associative memory model or a modern Hopfield network. This design demonstrably minimizes the energy functional that is tailored, enabling efficient retention of historical context while assimilating the incoming data. The efficacy of the model is comprehensively validated across a diverse range of temporal graph datasets, including those with long-range dependencies, demonstrating superior performance in both transductive and inductive scenarios for dynamic link prediction.

## 1 Introduction

Dynamical systems characterize the temporal evolution of complex phenomena across diverse domains from celestial mechanics to population dynamics (Strogatz, 2024). Traditionally, the modelling of dynamical systems relied on physics-based formulations derived from first principles, such as Newton's laws and conservation equations, where governing equations were explicitly obtained from fundamental physical insights. In modern settings, however, many real-world systems exhibit complex, nonlinear, and high-dimensional behaviours for which the exact dynamics are unknown or analytically intractable. In such cases, the governing function is derived directly from observational or experimental data (Brunton & Kutz, 2019). This data-driven approach faces significant challenges, including nonlinearities, chaotic regimes, transient phenomena, noise, stochastic effects, multi-scale interactions, and inherent uncertainties. Machine learning (ML) and deep learning (DL) methods have emerged as powerful tools to infer these dynamics from data, offering flexibility in capturing intricate patterns. Yet, to enhance interpretability, generalizability, and physical consistency, it is often beneficial to integrate domain knowledge —embedding known physics, symmetries, and constraints — into the learning framework, thereby combining the predictive strengths of data-driven models

with the robustness of physics-based reasoning. Amongst such hybrid approaches, energy-based models have emerged as a path to synergize data-driven learning with principled notions of energy in dynamical systems.

Parallel to these developments, the field of Graph Neural Networks (GNNs) has advanced rapidly, enabling effective representation learning on relational data (Graziani et al., 2024). While the majority of existing research pertains to static graphs, many real-world graphs are dynamic in nature. This has motivated the emergence of temporal graph learning methodologies (Trivedi et al., 2018; Kumar et al., 2019; Rossi et al., 2020; Xu et al., 2020; Cong et al., 2023). Despite these advances, effectively managing information propagation across temporal and spatial dimensions remains a persistent challenge (Longa et al., 2023). Recent studies, such as that by Yu et al. (2023), have explored Transformer-based architectures (Vaswani et al., 2017) to better preserve long-range temporal information. However, such approaches often entail quadratic complexity inherent to attention mechanisms and lack strong theoretical foundations. More recently, Gravina et al. (2024) presented deep graph networks specifically tailored for CTDGs, offering theoretical guarantees to mitigate information loss. Nonetheless, despite their theoretical rigor, these models exhibit inconsistent empirical performance across standard temporal graph benchmarks. In this scenario, energy-based models could be seen as an alternative approach to the modelling of temporal graphs.

Concurrently, Associative Memory models, commonly referred to as Hopfield Networks (Hopfield, 1982; 1984b), have seen a resurgence, due to new theoretical insights on their memory capacity and novel architectural extensions (Chaudhry et al., 2023). Modern variants, termed Dense Associative Memories or Modern Hopfield Networks, leverage sharpened activation functions to amplify their memory storage capacities, achieving super-linear (Krotov & Hopfield, 2016) and even exponential growth (Demircigil et al., 2017). These advances position them as powerful mechanisms for structured information retrieval in machine learning tasks. Additionally, Demircigil et al. (2017) highlights a compelling connection between these networks and the attention mechanism in transformers, identifying the transformer attention as a special instance of Modern Hopfield Networks characterized by a softmax activation function. Moreover, prior works have explored energy-based approaches for static graph machine learning using modern Hopfield networks (Hoover et al., 2024; Ramsauer et al., 2020). Despite these advances, the potential of energy-based architectures and Hopfield-inspired memory mechanisms for temporal graph machine learning remains largely underexplored.

Motivated by the aforementioned challenges and observations, our work proposes a novel architecture rooted in the principles of energy-based models — the **Temporal Energy Transformer (TET)**— for temporal graph machine learning. Our framework builds upon the foundational components of the Temporal Graph Network (TGN) (Rossi et al., 2020), incorporating key modules such as memory, message module and embedding module. The primary innovation is a novel graph embedding module designed as a sequence of energy-based attention layers, wherein the model parameters are optimized through a newly designed energy functional tailored specifically for temporal graph learning. Such an energy-based formulation provides a principled way to enforce long-term information retention, which is a key limitation of previous approaches. Designing an appropriate energy functional is a non-trivial task, giving rise to a fundamental challenge: *How can energy-based models be extended to effectively propagate information across both temporal and spatial dimensions in dynamically varying graphs?* To address this, we design the energy functional by appropriately incorporating the information from a current stream of events with the information from the past events retained in the model's so-called 'memory'. The contributions of this paper are outlined as follows:

1. We present the first energy-based model for temporal graph machine learning `TETN` operating on continuous-time dynamic graphs represented as a sequence of events.

2. *Empirical findings:* We demonstrate that `TETN` achieves state-of-the-art performance across multiple temporal graph datasets, consistently outperforming prior methods in both transductive and inductive link prediction settings.

## 2 Preliminaries

**Dynamic graphs.** A dynamic graph (also called a temporal graph) is defined as a tuple of the form $\mathcal{G}(t) = \{\mathcal{V}(t), \mathcal{E}(t), \mathbf{X}(t), \mathbf{E}(t)\}, t \geq 0$ where $\mathcal{V}(t)$ is the set of all nodes of the graph and $\mathcal{E}(t)$ is the set of all

edges in the graph at time $t$. The matrices $\mathbf{X}(t)$ and $\mathbf{E}(t)$ are the node and edge feature matrices at time $t$. The way in which we observe a system of interacting entities (as a dynamic graph) distinguishes the type of dynamic graph as *discrete-time dynamic graph* (DTDG) or *continuous-time dynamic graph* (CTDG).

A DTDG, $\mathcal{G} = \{\mathcal{G}_t | t \in [t_0, t_n]\}$, consists of a sequence of static graphs, known as snapshots, observed at periodic time intervals. Here, $\mathcal{G}_t = (\mathcal{V}, \mathcal{E}, \mathbf{X}_t, \mathbf{E}_t)$ represents the graph observed at a particular timestamp, say at $t = T$ and we denote $\eta_u(T) = \{v : (u, v) \in \mathcal{E}(T)\}$ as the temporal neighbourhood of node $u$ and $\eta_u^K(T)$ as the $K$-hop temporal neighbourhood of node $u$.

In contrast, CTDG is a continuous stream of events observed over time, that is, $\mathcal{G} = \{o_t | t \in [t_0, t_n]\}$, with events occurring at any time (irregular timestamps). Events are of three types: *node-wise* events wherein a node is created; *edge-wise* events wherein a temporal edge is created; *deletion* events when a node/edge is deleted. And, we define the temporal neighbourhood of node $u$ as $\eta_u([0, t]) = \{v : (u, v) \in o_t, t \in [0, t]\}$.

**Problem Definition.** Let $\{o_{t_1}, o_{t_2}, \ldots, o_{t_n}\}$ denote a sequence of timestamped events occurring at nodes $u \in \mathcal{V}(t)$ within a continuous-time dynamic graph (CTDG). We seek to develop a representation learning framework with an energy-based embedding module that ensures long-term retention and propagation of event information across the graph. Concretely, for any node $u$, if an event $o_{t_i}$ occurs at time $t_i$, the embedding $h_u(t)$ at any subsequent time $t > t_i$ must satisfy the condition: $h_u(t)$ retains information about event $o_{t_i}$, $\forall t > t_i$.

## 3 Temporal Energy Transformer (`TET`)

In this work, we propose a **Temporal Energy Transformer (`TET`)**, building upon the principles of conventional Energy Transformers, `ET` (Hoover et al., 2024), to address the challenges of temporal graph learning. While the existing model for static graph machine learning, `ET`, minimizes an energy function to align node representations with its immediate neighbours, our `TET` model is specifically tailored to incorporate both current and historical graph dynamics into the energy function. This is essential for integration into existing frameworks for temporal graph machine learning, such as Temporal graph Networks (TGN) (Rossi et al., 2020). Existing attention-based models, such as DyGFormer (Yu et al., 2023), often struggle to retain long-term historical information in dynamic graph tasks (Gravina et al., 2024). Our proposed TET model directly addresses this limitation by appropriately fusing past information into the learning process. Building upon the Energy Transformer framework (Hoover et al., 2024), our approach integrates two energy components: Energy attention and Hopfield-inspired energy, which we describe in detail in Section 5. The Energy attention component computes attention coefficients for a node's neighbours, akin to the Graph Attention Network (GAT) framework. The Hopfield energy, whose equivalence to the Transformer architecture (Vaswani et al., 2017) has been demonstrated (Ramsauer et al., 2020), is central to our model's capacity for long-range information retention.

Let us consider a CTDG graph $\mathcal{G}$. For a given node $u$ at time $t$, the model processes $x_u(t)$, which is the information of node $u$ from the current batch of events (e.g., recent edge additions), and $\eta_u^K([0, t])$, which captures its $K$-hop temporal neighbourhood spanning all past interactions up to time $t$.

The core of our model's learning process is to minimize the energy functional $E(t)$, defined for a stream of events at time $t$ as:

$$E(t) = \sum_{u=1}^{|\mathcal{V}(t)|} \sum_{v \in \eta_u^K([0,t])} f(x_u(t), s_u(t), e_{uv}(t)). \tag{1}$$

Here, $f(\cdot)$ is a learnable function that quantifies the interaction potential or compatibility between the current representation of node $u$, $x_u(t)$, its compressed representation in memory, denoted by $s_u(t)$ and the representation of edges, $e_{uv}(t)$ in its $K$-hop temporal neighbourhood, $\eta_u^K([0, t])$. $s_u(t)$ denotes a concise, aggregated representation of node $u$'s historical interactions. The total energy of the CTDG system is then $E = \sum_{t < T} E(t)$. As a consequence of the learning, minimizing $E(t) \forall t$ realizes a gradual decrease in $E$ over time thereby ensuring that the whole system moves towards a low energy configuration, though it may not be equivalent to the global minima (or a fixed attractor state). A salient property of an energy-based model is that learning is guided by the minimization of the function $f$ and consequently the functional $E$ which encapsulates the system's configuration. In contrast to a general loss-based learning

framework, which quantifies the discrepancy between predicted and true class probabilities for discrete outcomes; the energy-based minimization ensures that a continuous measure of the inherent dynamics of the system evolves smoothly towards a low-energy configuration (Lecun et al., 2006). To the best of our knowledge, our model is the first to introduce such an energy-based approach for temporal graph machine learning.

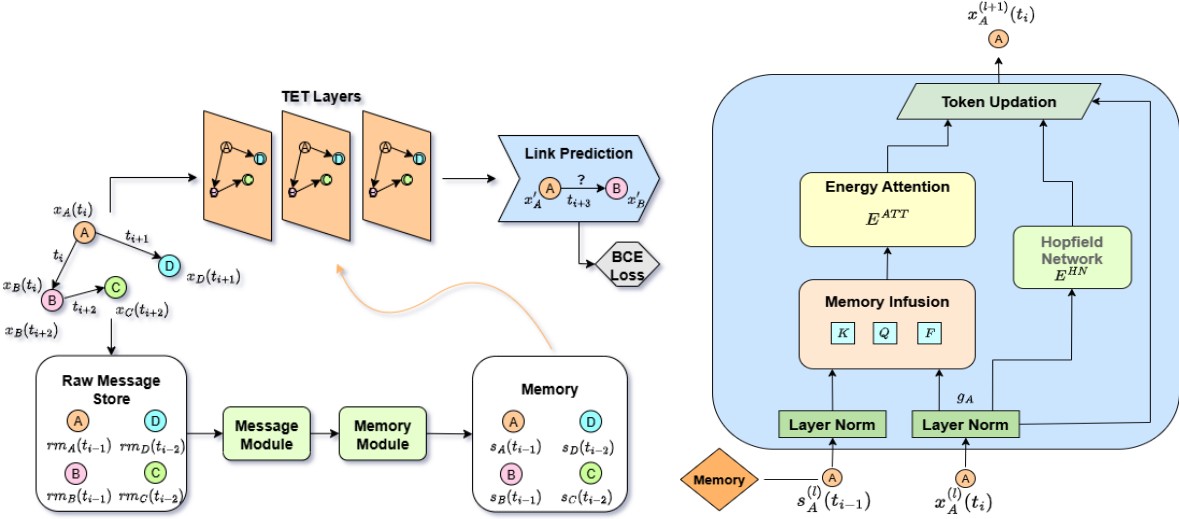

Figure 1: **TETN** pipeline consisting of the message module, memory and the graph propagation module (TET).

Figure 2: The block diagram represents the **Temporal Energy Transformer (TET)** layer in the architecture of Figure 1, showcasing how the embedding of a node A gets updated.

## 4 Energy-based framework for Temporal Graph Machine Learning

We describe below the proposed **TETN** framework, incorporating an energy-based graph propagation module into the TGN framework for CTDGs (Rossi et al., 2020). This enhanced framework is structured around its fundamental components: **(1) the Message module**, **(2) Memory**, and the **(3) Graph Propagation module**. Figure 1 provides a detailed depiction of the complete architecture. *The core contribution of this work lies in the design of our novel energy-based graph propagation module depicted in Figure 2, with detailed training and inference procedures outlined in Algorithm 1 and Algorithm 2 in Appendix A.12.* Additionally, the story of one CTDG event flowing through our network is provided in Appendix A.3.

To formally define the operations in our framework, we first establish the notation for this setting. Consider a CTDG graph $\mathcal{G}$ described by a stream of events occurring in the time interval $[0, T]$. Let the nodes in our graph be indexed by $A, B, C, \ldots$ and edges be represented as $AB, BC, \ldots$. We denote the corresponding node embeddings at a time instant $t$ as $x_A(t), x_B(t), \ldots, x_N(t)$ and edge embeddings as $e_{AB}(t), e_{BC}(t), \ldots$. Furthermore, the memory states of these nodes are represented by $s_A(t), s_B(t), s_C(t), \ldots s_N(t)$. Then $\mathcal{G}_t = (V(t), \mathcal{E}(t))$, where $V(t) = \{k : \exists x_k(t_i), t_i \in [0, t]\}$, $\mathcal{E}(t) = \{(i, j) : \exists e_{ij}(t_i), t_i \in [0, t]\}$. For simplicity, we describe the energy $E(t)$ as $E$ in the subsequent discussions. With these established notations, we describe the main modules of the framework below.

### 4.1 Message Module.

The message module is responsible for generating messages that facilitate the update of nodes' memory states within the temporal graph network. Specifically, whenever a new interaction event occurs at time $t$, messages are computed for both the source node $A$ and the target node $B$. These messages encapsulate

information pertinent to the current interaction and the historical state of the nodes involved. For a source node $A$ and a target node $B$, their respective messages, $m_A(t)$ and $m_B(t)$, are formulated as follows:

$$m_A(t) = \mathrm{msg}(rm_A(t^-)), \ m_B(t) = \mathrm{msg}(rm_B(t^-)), \tag{2}$$

where the raw messages, as stored in the raw message store, $rm_A(t)$ and $rm_B(t)$, are updated at end of the forward pass as:

$$rm_A(t) = s_A(t^-) \,\|\, s_B(t^-) \,\|\, \theta(t - t_A) \,\|\, e_{AB}(t),$$
$$rm_B(t) = s_B(t^-) \,\|\, s_A(t^-) \,\|\, \theta(t - t_B) \,\|\, e_{AB}(t).$$

In these equations, $s_A(t^-)$ and $s_B(t^-)$ represent the memory vectors of nodes $A$ and $B$, respectively, computed during their last update, prior to the current event. The terms $t_A$ and $t_B$ denote the timestamps of the most recent interactions involving nodes $A$ and $B$. The function $\theta(\cdot)$ processes the time difference between the current event and the node's last interaction and provides a functional time encoding (Xu et al., 2020). $\mathbf{e}_{AB}(t)$ represents features associated with the edge between $A$ and $B$ at time $t$. The operator $\|$ signifies vector concatenation. The function $\mathrm{msg}(\cdot)$ is a learnable function. In scenarios where multiple events involving the same node occur within the same batch, the corresponding individual messages are aggregated into a single comprehensive message. This aggregation is performed using an aggregation function, denoted as agg:

$$\overline{m}_w(t) = \mathrm{agg}(m_w(t_1), \ldots, m_w(t_b)),$$

where $w$ refers to the node (either $A$ or $B$), and $\mathrm{agg}(\cdot)$ can be implemented using various strategies, such as mean, sum, or simply taking the last message.

## 4.2 Memory.

The memory module serves as a crucial component for maintaining and evolving the historical representations of individual nodes. Following each interaction event, a node's memory state is updated by integrating its prior memory with incoming messages generated by the Message Module. Specifically, given the aggregated incoming message $\bar{m}_A(t)$ for node $A$ at time $t$ and its memory state $s_A(t^-)$ immediately preceding this event, the updated memory representation, $s_A(t)$, is computed as:

$$s_A(t) = \mathrm{mem}(\overline{m}_A(t), s_A(t^-)).$$

The function $\mathrm{mem}(\cdot)$ typically employs an RNN-based architecture, such as a Gated Recurrent Unit (GRU) (Cho et al., 2014).

## 4.3 Graph Propagation Module.

The graph propagation module generates the temporal embeddings for the nodes involved in the current input batch of events. Once a node's memory is updated with new event information from the memory updater module, its representation is enriched through this module, that involves multiple (TET) layers. The input involves the node embeddings in the current batch of CTDG events between times $t_i$ and $t_{i+k}$, the temporal $K$-hop neighbourhood ($\mathcal{N}_A^K([0, t_{i+k}])$) of the involved nodes and their compressed representation $s_A(t)$ from the memory module. Note that only the nodes involved in the current batch of input events are updated during a forward pass through the TET layer.

In the following section, we describe our graph propagation approach built upon modifications of the energy transformer block of Hoover et al. (2024) (described in Appendix A.14). The general architectural framework of our TET block is visually represented in Figure 2. It's core components are described below.

### 4.3.1 Layer Norm.

Given a node embedding $x_A$, a layer-normalized representation $g_A$ is obtained using this layer.

$$g_{iA} = \gamma \frac{(x_{iA} - \bar{x}_A)}{\sqrt{\frac{1}{D}(x_{iA} - \bar{x}_A)^2 + \epsilon}} \text{ where } \bar{x}_A = \sum_{i=1}^{D} x_{iA}. \tag{3}$$

### 4.3.2  Memory Infusion.

The layer-normalized embeddings $g_A$ of the nodes and their corresponding memory states $s_A$ from the memory module are used to compute the key, query and edge tensors for the energy terms in `TETN`. They are operated upon by summation or concatenation like in Eqn. (7).

### 4.3.3  Energy Attention.

The energy attention $E^{ATT}$ is designed by modifying the conventional transformer-based attention mechanism, and is used to align the representation of a particular node with respect to the information of its neighbours. The memory-infused keys and queries (and/or edge tensors) for each node are used here to evaluate the attention. The model leverages memory-infused tensors for cross-temporal attention, thereby aligning node representations across time. The exact formulation for the attention based energy is provided in Eqn. (9).

### 4.3.4  Hopfield Network.

The Hopfield Network (HN) aligns the node representations to be consistent with the information learnt from the evolving graph across batches. This is in contrast to the HN block in `ET` (see A.13) which learns the patterns over the entire static graph. In `TETN`, the Hopfield energy is defined similarly to the one in `ET`:

$$E^{HN} = -\sum_{B=1}^{N} \sum_{\mu=1}^{K} G\left(\sum_{j} \xi_{\mu j} g_{jB}\right), \xi \in \mathbf{R}^{K \times D}, \tag{4}$$

where, $\xi_{\mu j}$ is a set of learnable weights, also called memories in HN, and $G(.)$ is an integral of the activation function $r(.)$. The activation function that we use is $r(.) = ReLU$.

### 4.3.5  Token Update.

The inference pass of the `TET` block is designed as an ODE update:

$$\tau \frac{dx_{iA}}{dt} = \mathcal{F}\left(\frac{\partial E}{\partial g}, \frac{\partial E}{\partial s}, \frac{\partial E}{\partial e}\right), \tag{5}$$

where, $E = E^{ATT} + E^{HN}$. Here, $E$ is total energy, $E^{ATT}$ is the attention based energy term and $E^{HN}$ is the contribution to energy from Hopfield network. The exact formulation of the update function $\mathcal{F}$ for `TETN` is described in Eqn. (11).

## 5  Energy in Temporal Energy Transformer (`TET`)

In this section, we describe in detail the energy formulation in Temporal Energy Transformer (`TET`). The energy formulation used in ET (Hoover et al., 2024) does not explicitly use the edge features and is limited to learning from static graphs. Our `TETN` model, designed for temporal learning, involves a modified energy attention with cross-temporal attention terms to align the node representations across the temporal neighbourhood. Moreover, the edge features are added to the computation of the key tensor with an additional learnable matrix $W^F \in \mathbf{R}^{Y \times H \times D}$. Further to this, the memory infusion module involves the consideration of the memory states of the nodes $s_A(t)$ in the computation of the new key, query and edge tensors as:

$$K_{\alpha hB} = \sum_{j} W^K_{\alpha hj}(g_{jB} + s_{jB}), K \in \mathbf{R}^{Y \times H \times N}, \tag{6}$$

$$Q_{\alpha hC} = \sum_{j} W^Q_{\alpha hj}(g_{jC} + s_{jC}), Q \in \mathbf{R}^{Y \times H \times N}, \tag{7}$$

$$F_{\alpha hBC} = \sum_{j} W^F_{\alpha hj} e_{jBC}, F \in \mathbf{R}^{Y \times H \times N}. \tag{8}$$

Incorporating the above modifications into the classical energy attention equation of ET (Hoover et al., 2024) (see A.14), we have the modified energy attention equation given as:

$$E^{ATT} = -\frac{1}{\beta} \sum_{h=1}^{H} \sum_{C=1}^{N} log \left( \sum_{B \in \mathcal{N}_C} \exp A_{hBC} \right), \tag{9}$$

where, $A_{hBC} = \beta \sum_\alpha (K_{\alpha hB} + F_{\alpha hBC}) Q_{\alpha hC}$. Combining $E^{ATT}$ with the Hopfield Energy from Eqn. (4), we have the total energy as:

$$E = E^{ATT} + E^{HN} \tag{10}$$

A toy example involving three nodes is described to intuitively explain the modified energy function, in Appendix A.2. While in the ET, the energy function depends solely on the node embeddings $g$ of the current input stream of events, the proposed energy functional in `TETN` depends not only on the current node embeddings $g$, but also their corresponding memory states $s$ and the features $e$ of the involved edges. Consequently, the token update for each forward pass must take into consideration the variations of energy $E$ with respect to each of these variables involved. Specifically, the ODE-based update for $x$ is guided by the gradient $\frac{\partial E}{\partial g_A}$, ensuring descent along the energy landscape towards a minimum. While the descent direction is definite, we drive the descent by defining an adaptive step length which incorporates a linear combination of the energy variation with respect to $s$ and $e$ as given below. This integration preserves the core principle that token updates should reduce energy (see proof of Theorem 1 in Appendix A.1), while simultaneously supporting stable and consistent learning dynamics. Thus, the token update, following Eqn. (5), is defined as:

$$\tau \frac{dx_{iA}}{dt} = -\frac{\partial E}{\partial g_{iA}} \left( m + \gamma_1 \left\| \frac{\partial E}{\partial s_{iA}} \right\| + \gamma_2 \sum_{C \in \mathcal{N}_A} \left\| \frac{\partial E}{\partial e_{iCA}} \right\| \right). \tag{11}$$

where $\gamma_1$ and $\gamma_2$ are hyperparameters and $m$ is a margin whose default value is 0 and it takes the value 1, if $\gamma_1, \gamma_2 = 0$. The token update process defined by Eqn. (11) integrates three terms: $\frac{\partial E}{\partial g_{iA}}$ incorporates information from the **current batch**, $\frac{\partial E}{\partial s_{iA}}$ leverages data from **memory**, and $\frac{\partial E}{\partial e_{iCA}}$ captures **edge-specific information**, as explained above. Since we update only the current node embeddings in the `TETN` layers, we do not consider layer-wise update equations for $s$ or $e$. Here, the energy gradients (assuming only one head of attention) are given as:

$$-\frac{\partial E^{ATT}}{\partial g_{iA}} = -\frac{\partial E^{ATT}}{\partial s_{iA}} = \sum_{C \in \mathcal{N}_A} \sum_\alpha W_{\alpha i}^Q (K_{\alpha C} + F_{\alpha AC}) \omega_{CA} + W_{\alpha i}^K Q_{\alpha C} \omega_{AC}$$

$$-\frac{\partial E^{ATT}}{\partial e_{iCA}} = \sum_\alpha W_{\alpha i}^F Q_{\alpha A} \omega_{CA}$$

where,

$$\omega_{CA} = \underset{C}{\mathrm{softmax}} \left( \beta \sum_\gamma (K_{\gamma C} + F_{\gamma AC}) Q_{\gamma A} \right) \text{ and } \omega_{AC} = \underset{A}{\mathrm{softmax}} \left( \beta \sum_\gamma (K_{\gamma A} + F_{\gamma AA}) Q_{\gamma C} \right).$$

## 6 Experiments

We assess our model's capability on temporal representation learning by evaluating it on the CTDG benchmark datasets. We explain the experimental setup in more detail in Section 6.1 and discuss the results in Section 6.2. Additionally, in Appendix A.1, we demonstrate the provable, monotonic decrease of our energy functional over time. We present brief descriptions and statistics of the datasets in Appendix A.10.

Table 1: Results of the future link prediction task - transductive setting. We report the mean test set average precision (AP) and standard deviation in percent averaged over random weight initializations. Methods are ranked by average rank across all datasets (lower is better).

| Dataset → / Method ↓ | Wikipedia | UCI | Reddit | MOOC | Enron | LastFM | Avg. Rank |
|---|---|---|---|---|---|---|---|
| **TETN (ours)** | $\mathbf{99.20}_{\pm 0.05}$ | $\mathbf{98.98}_{\pm 0.03}$ | $\mathbf{99.42}_{\pm 0.69}$ | $85.07_{\pm 1.10}$ | $\mathbf{92.67}_{\pm 0.04}$ | $\mathbf{96.68}_{\pm 0.14}$ | **1.67** |
| DyGMamba | $99.15_{\pm 0.02}$ | $95.91_{\pm 0.15}$ | $99.32_{\pm 0.01}$ | $\mathbf{89.21}_{\pm 0.08}$ | $92.65_{\pm 0.12}$ | $93.35_{\pm 0.20}$ | 1.83 |
| DyGFormer | $99.03_{\pm 0.03}$ | $95.74_{\pm 0.17}$ | $99.22_{\pm 0.01}$ | $87.66_{\pm 0.48}$ | $92.42_{\pm 0.11}$ | $92.95_{\pm 0.14}$ | 3.00 |
| TGN | $98.45_{\pm 0.01}$ | $92.33_{\pm 0.64}$ | $98.65_{\pm 0.04}$ | $89.15_{\pm 1.69}$ | $86.98_{\pm 1.05}$ | $75.31_{\pm 5.62}$ | 4.33 |
| GraphMixer | $97.22_{\pm 0.02}$ | $93.15_{\pm 0.41}$ | $97.31_{\pm 0.01}$ | $82.80_{\pm 0.15}$ | $82.13_{\pm 0.30}$ | $75.56_{\pm 0.19}$ | 5.50 |
| CTAN | $96.61_{\pm 0.79}$ | $76.64_{\pm 4.11}$ | $97.21_{\pm 0.84}$ | $84.71_{\pm 2.85}$ | $92.52_{\pm 1.20}$ | $86.44_{\pm 0.80}$ | 6.17 |
| TGAT | $96.88_{\pm 0.06}$ | $79.40_{\pm 0.61}$ | $98.57_{\pm 0.01}$ | $85.71_{\pm 0.20}$ | $70.76_{\pm 1.05}$ | $73.30_{\pm 0.18}$ | 6.50 |
| JODIE | $96.51_{\pm 0.22}$ | $89.28_{\pm 1.02}$ | $98.31_{\pm 0.06}$ | $81.04_{\pm 0.83}$ | $84.85_{\pm 3.13}$ | $70.95_{\pm 2.94}$ | 7.83 |
| DyRep | $94.88_{\pm 0.29}$ | $66.11_{\pm 2.75}$ | $98.18_{\pm 0.03}$ | $81.50_{\pm 0.77}$ | $79.80_{\pm 2.28}$ | $71.85_{\pm 2.44}$ | 8.67 |

Table 2: Results of the future link prediction task - inductive setting. We report the mean test set average precision (AP) and standard deviation in percent averaged over random weight initializations. Methods are ranked by average rank across all datasets (lower is better).

| Dataset → / Method ↓ | Wikipedia | UCI | Reddit | MOOC | Enron | LastFM | Avg. Rank |
|---|---|---|---|---|---|---|---|
| DyGMamba | $98.77_{\pm 0.03}$ | $94.76_{\pm 0.19}$ | $98.97_{\pm 0.01}$ | $\mathbf{88.64}_{\pm 0.08}$ | $\mathbf{89.67}_{\pm 0.27}$ | $94.42_{\pm 0.21}$ | **1.67** |
| **TETN (ours)** | $\mathbf{98.83}_{\pm 0.24}$ | $\mathbf{95.44}_{\pm 0.15}$ | $\mathbf{99.01}_{\pm 0.24}$ | $80.87_{\pm 0.14}$ | $89.65_{\pm 0.22}$ | $\mathbf{97.69}_{\pm 0.21}$ | 2.00 |
| DyGFormer | $98.58_{\pm 0.01}$ | $94.45_{\pm 0.13}$ | $98.83_{\pm 0.02}$ | $87.05_{\pm 0.51}$ | $89.62_{\pm 0.27}$ | $94.17_{\pm 0.10}$ | 3.00 |
| TGN | $97.81_{\pm 0.18}$ | $87.81_{\pm 1.32}$ | $97.41_{\pm 0.12}$ | $88.01_{\pm 1.48}$ | $78.76_{\pm 1.69}$ | $81.18_{\pm 3.27}$ | 4.67 |
| GraphMixer | $96.61_{\pm 0.04}$ | $91.17_{\pm 0.29}$ | $95.24_{\pm 0.08}$ | $81.38_{\pm 0.17}$ | $75.55_{\pm 0.81}$ | $82.07_{\pm 0.31}$ | 6.00 |
| JODIE | $94.91_{\pm 0.32}$ | $79.73_{\pm 1.48}$ | $96.43_{\pm 0.16}$ | $80.57_{\pm 0.52}$ | $78.97_{\pm 1.59}$ | $83.13_{\pm 1.19}$ | 6.00 |
| TGAT | $96.26_{\pm 0.12}$ | $79.10_{\pm 0.49}$ | $97.13_{\pm 0.04}$ | $85.28_{\pm 0.30}$ | $66.67_{\pm 1.07}$ | $78.40_{\pm 0.30}$ | 6.50 |
| DyRep | $92.21_{\pm 0.29}$ | $58.39_{\pm 2.38}$ | $95.89_{\pm 0.26}$ | $80.50_{\pm 0.68}$ | $73.97_{\pm 3.00}$ | $83.47_{\pm 1.06}$ | 7.33 |
| CTAN | $93.58_{\pm 0.65}$ | $49.78_{\pm 5.02}$ | $80.07_{\pm 2.53}$ | $64.99_{\pm 2.24}$ | $74.61_{\pm 1.64}$ | $60.40_{\pm 3.01}$ | 8.50 |

## 6.1 Experimental Setup

**Datasets:** We test the performance of `TETN` on CTDG learning task by comparing it with other baseline methods on dynamic link prediction task on CTDG datasets. We consider six real world datasets collected by Poursafaei et al. (2022), namely, Wikipedia, UCI, Reddit, MOOC, Enron, LastFM (see Table 11). These datasets are very widely used in some recent papers like Ding et al. (2024) and Yu et al. (2023). Out of these datasets MOOC, Enron and LastFM are considered long range temporal dependent datasets as established by Yu et al. (2023).

**Baselines:** We compare `TETN` with eight recent CTDG models, viz., DyRep (Trivedi et al., 2018), JODIE (Kumar et al., 2019), TGAT (Xu et al., 2020), TGN (Rossi et al., 2020), GraphMixer (Cong et al., 2023), CTAN (Gravina et al., 2024), DyGFormer (Yu et al., 2023) and DyGMamba (Ding et al., 2024). Among these models, CTAN, DyGFormer and DyGMamba are designed to capture long-range temporal information propagation and we compare them with `TETN` and highlight the advantage of using energy based models.

**Implementation Details:** We follow the experimental protocol established by Ding et al. (2024) to ensure fair comparison with existing baselines. Our evaluation employs the same datasets, metrics (average precision (AP) and area under the Receiver Operating Characteristic curve (AUC-ROC)), and experimental configurations as the comparative methods, maintaining consistency across all benchmarks to eliminate potential evaluation bias. The code for model and experiments are present at `https://github.com/ast-fri/temporal-energy-transformer`.

**Evaluation:** We employ two evaluation settings following the previous works: the transductive and inductive settings. We use a random negative sampling strategy for inductive and transductive setting. Model performance is evaluated using the metrics, AP and AUC-ROC. All experiments are implemented in PyTorch and on a server with NVIDIA A30 GPUs with 24GB of RAM. To ensure statistical reliability, all experiments were conducted across five independent runs with different random initializations. Tables 1, 2, 3 and 4 report the mean performance along with standard deviations. We also do hyper-parameter tuning for `TETN` and the details are mentioned in Appendix A.11.

## 6.2 Performance of `TETN`

We present the future link prediction results for transductive and inductive settings in Tables 1 and 2, respectively. Across both these evaluations, `TETN` demonstrates consistently strong performance, establishing itself as a robust and generalizable approach for future link prediction in dynamic graphs. In the **transductive setting**, `TETN` attains the best performance on majority of the datasets, achieving an average rank of **1.67**. It secures top performance on UCI (98.98% AP) and LastFM (96.68% AP) datasets showcasing an improvement in the metric with high margins. And it maintains highly competitive results on remaining datasets with performance improvement within a narrow margin of the leading methods. The performance on the long-range datasets (Yu et al., 2023), particularly *Enron* and *LastFM*, further pronounces `TETN`'s robustness. *LastFM* is a dense dataset spanning over a month with around 1.3 million events and *Enron* is a dataset spanning over 3 years with over 125*k* events (see Table 11 in Appendix A.10). The ability of `TETN` to maintain stable and expressive temporal embeddings under such diverse temporal settings indicates that its temporal encoding strategy preserves long-range temporal structure more effectively than attention-based or memory-based baselines. Importantly, `TETN` avoids common pitfalls such as non-dissipativeness over time and space (Gravina et al., 2024), which degrades performance in continuous-time architectures. This robustness stems from `TETN`'s energy-based design.

In the **inductive setting**, where the model must generalize to previously unseen nodes, `TETN` achieves an average rank of **2.0**, closely following DyGMamba while outperforming transformer-based, recurrent, and structural baselines by a comfortable margin. Notably, `TETN` secures the best performance on datasets such as *Wikipedia*, *UCI*, *Reddit*, and *LastFM*, demonstrating that its learned temporal representations remain transferable even when the structural context shifts at test time. This capability is crucial for real-world dynamic graph applications—including recommender systems, communication platforms, and evolving knowledge graphs—where new entities frequently appear.

Performance on irregular datasets such as *MOOC* and *Enron* shows a noticeable sensitivity in the inductive setting. On *Enron*, the scores remain close to those of DyGFormer and DyGMamba, whereas on *MOOC* the gap widens. Unlike LastFM, which exhibits fine-grained and regular temporal evolution, Enron and MOOC suffer from limited temporal resolution, with many interactions collapsing onto the same timestamps, which might be diminishing the performance on these datasets.

Further to the AP results, we report the AUC-ROC results on the CTDG datasets in Tables 3 (transductive) and 4 (inductive) in Appendix A.4. We observe that the performance of the `TETN` model is in alignment with its performance based on the AP metric. We also discuss the statistical significance scores (see Table 5), for these performance metrics, for the datasets with overlapping performance with baselines, in Appendix A.5.

Additionally, we conducted auxiliary experiments on the tgbl-review dataset, which has nearly 5 million events and is long-range in both time and space. We refer the reader to Table 6 in Appendix A.6 for the results. `TETN` achieved an MRR (Mean Reciprocal Rank) of 0.375, outperforming DyGFormer (0.224 MRR) and TGN (0.349 MRR). This represents a substantial relative improvement and further highlights the superior long-range temporal propagation capability of `TETN`.

We also provide the results for ablation of various modules of our architecture in Tables 7 and 8 in Appendix A.7. The results demonstrate that each component, viz., Energy Attention, HN and edge-features, contributes significantly to the observed performance of our model. Further to this, Table 9 in A.8 reports the model's performance under different choices of the hyperparameters $\gamma_1$ and $\gamma_2$ in Eq. 11. This illustrates the significance of the various energy gradient terms in Eqn. 11. Further details and discussion are provided in the respective appendix sections.

## 7    Discussion

The performance of `TETN` across both transductive and inductive settings highlight an important strength: **TETN does not overfit to the observed graph structure**. Instead, it learns temporal representations that encode both short-range interaction signals and long-horizon dynamics in a manner that generalizes across graph topologies. This balance is notably difficult to achieve, as many existing temporal graph neural networks rely heavily on localized structural patterns or explicit memory modules that fail to transfer well to unseen nodes. `TETN` achieves top performance on UCI (both settings), Wikipedia (both settings), Reddit (both settings), LastFM (both settings) and Enron (transductive) with only a minor decrease in the performance metrics for Enron (inductive). These results confirm that our energy-based formulation excels at the long-range propagation problem that motivates this work. That an energy-based approach ranks as the top performing method (or a close second rank) among 9 established methods, including those designed explicitly for long-range modelling (like CTAN and DyGMamba), represents a significant advancement in establishing a new research direction.

`TETN`'s exceptional performance stems from its energy-based architecture for temporal graph learning, demonstrating that this previously unexplored paradigm can compete with highly optimized Transformer or SSM-based methods (viz., DyGFormer, DyGMamba), while providing theoretical guarantees (Theorem 1 on monotonic energy decrease). The model compresses each node's historical information into compact representations through the energy functional, enabling effective modelling of temporal evolution without computational bottlenecks. Further, the learning proceeds through minimization of the energy functional, which depends explicitly on node representations rather than a loss based on predictions alone. We consider this energy-based minimization as a more effective approach since the energy values themselves are an indication of the system's state at each update, making it more interpretable. This decomposition allows practitioners to interpret low-energy configurations as states where nodes are temporally and structurally consistent with their neighbourhoods, beyond mere prediction accuracy. Moreover, the energy plots in Appendix A.1 demonstrate that our formulated energy functional exhibits smooth convergence behaviour throughout training on the Wikipedia dataset, ensuring stable and optimal temporal node embedding updates.

The energy-based approach proves particularly effective on challenging datasets like LastFM and tgbl-review, with extensive edge counts, where conventional methods struggle with scalability and long-term memory requirements. We attribute the performance gain of `TETN` to three major building blocks of it's architecture: (i) the ODE-based update rules, (ii) the dense associative memory inherited from Hopfield-style dynamics, and (iii) the energy-based attention. By embedding past information directly into the energy functional (memory infusion), the model achieves more reliable long-range propagation than purely attention-driven mechanisms. Compared to contemporary architectures, `TETN` achieves these improvements without introducing additional features (as in DyGMamba or DyGFormer) or stabilizing terms (as in CTAN). Moreover, `TETN` can handle concurrent edges, while DyGMamba can handle edges (or events) only sequentially. We acknowledge limitations in performance on the MOOC dataset. Concurrently, we emphasize that opening this methodological paradigm, achieving state-of-the-art results on long-range tasks, and providing theoretical foundations absent in competitors constitute substantial contributions beyond point-wise empirical comparisons.

Overall, the results demonstrate that **TETN achieves a rare combination of accuracy, training stability, and generalization**. Its consistently strong performance across datasets of varying scale, sparsity, and temporal characteristics underscores its suitability as a general-purpose framework for temporal graph learning. These findings position `TETN` as a compelling model for advancing dynamic graph representation learning, particularly for AI applications requiring reliable forecasting over evolving relational structures.

## 8    Related Work

**Long range Propagation for Temporal Graph Machine Learning:** Despite recent efforts, long-range information retention in temporal graph machine learning remains a significant challenge. The Continuous-time Graph Anti-symmetric Network (CTAN) (Gravina et al., 2024) pioneered an ODE-based approach that captures long-term temporal dependencies through non-dissipative information propagation across time and

space. Building on prior work that incorporated asymmetric terms to mitigate over-smoothing in static graphs (Gravina et al., 2022), CTAN demonstrates superior performance over existing state-of-the-art methods including TGN (Rossi et al., 2020) and DyGFormer (Yu et al., 2023). However, CTAN shows low performance metrics on temporal datasets due to its reliance on self-attention mechanisms. Existing approaches for dynamic graph modelling, such as DyGMamba (Ding et al., 2024) (based on SSMs) and DyGFormer (Yu et al., 2023) (based on Transformers), effectively model long-range dependencies, but are often constrained by quadratic computational complexity at high patch sizes. To overcome these limitations, we introduce a novel energy-based method. Our approach combines an **Energy attention block** and a **Hopfield Network** (Section 4.3), a fusion that offers a compelling alternative. Specifically, we leverage the Hopfield Network for its proven equivalence to the self-attention mechanism used in DyGFormer, as demonstrated in this work (Ramsauer et al., 2020). Critically, unlike the black-box nature of many Transformer-based models, our network's parameters are optimized using a curated energy function designed for long-range information retention, thereby enhancing model transparency and providing a path toward more explainable dynamic graph models.

**Temporal Graph Machine Learning:** The challenge of modelling evolving network structures has led to diverse methodological innovations in temporal graph analysis. Contemporary approaches utilize different architectural principles and temporal modelling strategies. Recurrent-based methods like JODIE (Kumar et al., 2019) employ sequential neural architectures to maintain evolving node representations through interaction history processing. GraphMixer (Cong et al., 2023) introduces a streamlined architecture with dedicated encoding components for nodes and edges, integrated with MLP-Mixer modules for feature synthesis. Hybrid approaches address both spatial and temporal complexities by integrating multiple modelling paradigms. TGAT (Xu et al., 2020) incorporates temporal encoding directly into node feature representations, while TGN (Rossi et al., 2020) combines recurrent processing with attention-based spatial modelling for comprehensive spatio-temporal learning. Our framework builds upon several core modules from TGN, with a key distinction: our graph propagation module incorporates an energy-based model designed using attention and Hopfield networks, which help preserve long-range information.

**Energy-based methods for deep learning:** Energy-based models (EBMs) provide a unified framework for learning by associating scalar energy values to variable configurations, where inference involves finding minimum energy states without probabilistic normalization (Lecun et al., 2006). Recent advances have leveraged EBMs for neural reasoning by parameterizing energy landscapes over output spaces, enabling adaptive computational allocation where harder problems receive more optimization steps (Du et al., 2022). Beyond reasoning, EBMs have also proven effective in generative modelling (Du & Mordatch, 2020; Grathwohl et al., 2020) and in biologically-motivated learning rules such as equilibrium propagation (Scellier & Bengio, 2017), demonstrating their versatility as a learning framework across diverse deep learning settings. Extending energy-based formulations to temporal graph machine learning, however, remains largely unexplored.

The Energy Transformer (Hoover et al., 2024) minimizes a global energy function $E$, with minima corresponding to fixed attractor states. While promising for static graphs, adapting energy functionals to temporal graphs and addressing long-range propagation challenges remain non-trivial. Our energy-based framework models interacting entities as a dynamical system of ODEs driven by an energy function that represents entity relationships, specifically considering the temporal setting. We introduce the Temporal Energy Transformer (TET) layer, a novel architecture designed for energy-based "temporal"graph machine learning. The TET layer has two novel aspects designed for capturing long-range dependencies in temporal graph machine learning: *memory infusion* (Section 4.3.2), which integrates each node's compressed historical interaction summary with its current batch representation to enable long-term temporal reasoning; and an *energy attention block* (Section 4.3.3), which employs a modified energy-based attention mechanism augmented with cross-temporal attention terms to align node representations across the temporal neighbourhood, in contrast to the attention terms in ET layers which only consider spatial neighbourhood. Together, these two components form a principled framework that, to the best of our knowledge, represents the first application of energy-based models to temporal graph machine learning.

**Oversquashing in static and dynamic graphs:** To mitigate the oversquashing problem, wherein deep graph neural networks struggle to capture long-range dependencies, several ODE-based methods have been proposed for static graphs (Chamberlain et al., 2021; Eliasof et al., 2021). Inspired by these developments, our

`TETN` token update rule is also formulated as an ODE. For discrete-time dynamic graphs (DTDGs), existing approaches such as Pareja et al. (2020) model graph evolution through recurrent architectures, while Ceni et al. (2025) employs graph state-space models to address information propagation loss. By contrast, our energy-based formulation naturally generalises to continuous-time dynamic graphs (CTDGs), which are a realistic representation of real-world networks.

## 9 Conclusion

In this work, we present Temporal Energy Transformer Network (`TETN`), as a first attempt of introducing energy-based optimization principles to continuous-time dynamic graph learning. Our framework fundamentally reconceptualizes temporal representation learning by formulating node evolution as an energy minimization problem, providing theoretical grounding for capturing complex temporal dynamics in networked systems. The proposed energy functional establishes a unified mathematical framework that naturally balances temporal continuity with structural coherence, addressing key limitations in existing temporal graph neural networks. Beyond demonstrating strong empirical performance, this work establishes energy-based methods as a viable paradigm for dynamic graph analysis. The theoretical foundation provided by our energy formulation opens avenues for incorporating first principles into graph learning, potentially enabling more interpretable approaches to modelling real-world dynamic systems. Moreover, as this work establishes the framework to use energy-based models for temporal graph learning, further improved designs of energy functionals while retaining the current framework could be potentially explored in future. We believe our work represents a significant step toward bridging physics-inspired optimization with modern graph neural network architectures.

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

## A  Appendix

### A.1  Analysis of Energy

One of the major considerations while designing an energy function whose gradient controls the dynamics of the neural ODE is that the energy function must be a monotonically decreasing function.

**Theorem 1.** *The energy functional $E$ in Eqn. (10) of* `TETN` *is a decreasing function under the dynamics defined by the ODE-update equation Eqn. (11), for $\beta, \gamma_1, \gamma_2 \in \mathbb{R}^+$ and $m = 0$.*

*Proof.* Consider energy $E$ of `TETN`. We have $E(t) = E(g_A, s_A, e_{AC} | A \in \mathcal{V}(t) \, \text{and} \, (A, C) \in \mathcal{E}(t))$ and the update from one layer to the next is given by $\frac{dE}{dt}$ as:

$$\frac{dE}{dt} = \sum_{i,j,A} \frac{\partial E}{\partial g_{iA}} \frac{\partial g_{iA}}{\partial x_{jA}} \frac{dx_{jA}}{dt} + \sum_{i,A} \frac{\partial E}{\partial s_{iA}} \frac{ds_{iA}}{dt} + \sum_{i,A,C} \frac{\partial E}{\partial e_{iAC}} \frac{de_{iAC}}{dt}. \tag{12}$$

Since, for a given pass through the `TETN` layer, the ODE system only updates the node features, while the memory states and the edge features remain unchanged, we get $\frac{ds_{iA}}{dt} = 0$ and $\frac{de_{iAC}}{dt} = 0$. Additionally, we have,

$$\mathbf{M}_{ij}^A = \frac{\partial g_{iA}}{\partial x_{jA}} = \frac{\partial^2 L}{\partial x_{iA} \partial x_{jA}},$$

where $L$ is the Lagrangian given as

$$L = D\gamma\sqrt{\frac{1}{D}(x_{iA} - \bar{x}_A)^2 + \epsilon} + \sum_j \delta_j x_j.$$

So, $\mathbf{M}$ is a positive semi-definite matrix. Hence, using Eqn. (11) with $m = 0$, we get,

$$\frac{dE}{dt} = -\sum_{i,j,A}\left(\frac{\partial E}{\partial g_{iA}}\mathbf{M}_{ij}^A\frac{\partial E}{\partial g_{jA}}P_{jA}\right) \leq 0,\tag{13}$$

since $P_{jA} = \left(\left\|\frac{\partial E}{\partial s_{jA}}\right\| + \sum_{C \in \mathcal{N}_A}\left\|\frac{\partial E}{\partial e_{jCA}}\right\|\right) \geq 0.$ $\qquad\square$

**Remark:** For $m = 1$ and $\gamma_1, \gamma_2 = 0$, Eqn. 11 reduces to

$$\tau\frac{dx_{iA}}{dt} = -\frac{\partial E}{\partial g_{iA}}$$

and $E$ can be proved to be a decreasing functional (proof similar as above).

Our experiments empirically demonstrate that energy is a decreasing function across the `TETN` layers. As shown in Figures 3 and 4, the layer-wise energy decreases as training progresses. Figure 3 plots the average energy per layer across training epochs. Initially, the energy values decrease and enter a stable, low-energy region (indicated by the colored area in the figures). As the training continues, the model begins to overfit the training data, causing the energy values to increase, a trend visible in Figure 3 after epoch 125. For a more fine-grained analysis, Figure 4 displays the energy at every gradient update (i.e., for each batch). This plot clearly shows that as the model processes more batches, the loss decreases, and the model's energy also decreases, demonstrating a synergistic relationship between learning and energy. We note that even with a dynamically evolving dataset, our energy-based model directs the system toward a basin of attraction and a local minimum, rather than continuously decreasing. This supports our hypothesis that the model learns a few global patterns in a dynamic dataset, allowing it to predict accurately even when some data properties change.

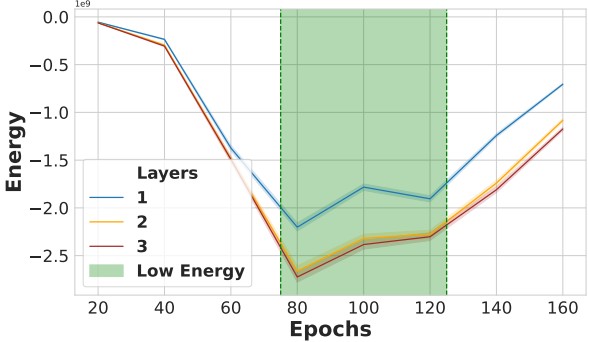
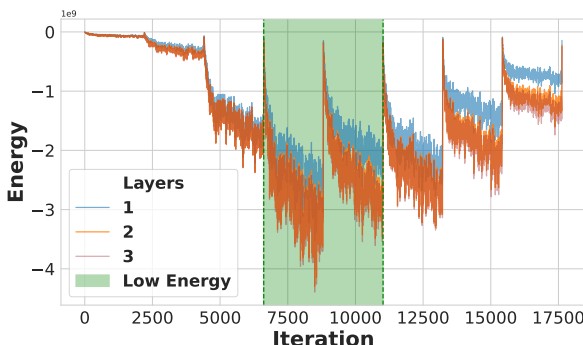

Figure 3: Energy over epochs across `TETN` layers | Figure 4: Energy over iterations across `TETN` layers.

We further explain the behavior of the energy curves in Figure 4. The oscillatory behavior in the energy curves during later iterations can be attributed to the non-i.i.d. nature of our training batches. In dynamic graph learning pipelines, batches are typically constructed by sorting edges chronologically based on their addition (event) timestamps. This temporal ordering means that consecutive batches can exhibit significantly different graph structures and patterns. As the energy function parameters are updated batch-wise, each update optimizes for the current batch's characteristics, which may slightly deviate from the optimal parameters for

earlier temporal segments of the graph. Consequently, at the beginning of each new epoch, when the model re-encounters the initial batches, the energy values start relatively high due to this parameter misalignment. However, as training progresses through the epoch and the model processes a few batches, the parameters quickly realign to better fit the early temporal patterns, causing the energy to decrease rapidly. This creates the oscillatory pattern observed in Figure 4. It is worth noting that Figure 3 demonstrates the overall decreasing trend in energy across epochs, capturing the global convergence behavior of our model. We emphasize that the local oscillations in Figure 4, while present, do not detract from the fundamental learning progress of the system.

## A.2 Energy Formulation: Intuition

**Objective and Setup:** Our objective is to provide an intuitive explanation for our energy formulation and explain how it differs from the energy in Energy Transformer method (Hoover et al., 2024). The example provided below helps us clearly state our advantage and highlight our contribution. Let us assume a setting with 3 nodes $A, B$, and $C$, where node $A$ and $B$ already have an edge between them, see Figure 5 (top). We represent the node features and memory states for these nodes as $g_i$ and $s_i$ respectively for $i \in \{A, B, C\}$. If these features are at a timestep $t$, then we could compute the features after message passing (i.e., next energy update) as $g_i^+$, using our energy update equation 11.

**Assumption:** We assume that the weights of the models are identity matrices and for any given node, the memory module stores only the node features associated with it's last event, for a simplified approach. Finally, we also ignore all continuous monotonic functions like log and exponent, which further simplifies the analysis. These assumptions help us pinpoint the exact advantage of our energy-based temporal formulation over standard energy-based methods. These assumptions allows us to use node features as node embeddings, before the message passing step using our energy functional.

**Task:** We consider a streaming setting where events arrive sequentially, and we assume a simple downstream task of link prediction. To predict whether a link exists between the nodes or not, we take the similarity of node embeddings and threshold it based on the dot product.

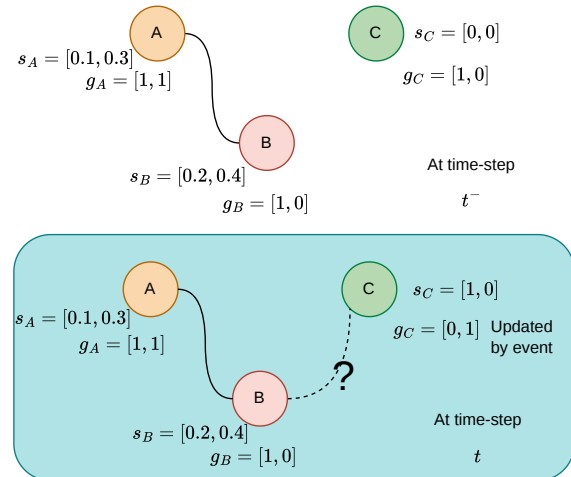

Figure 5: Toy example setup to show intuition of energy functional and the advantage of `TETN`.

**Example:** For the above setup, we now consider an example for a three-node graph as shown in Figure 5. Let us assume node features for the nodes are $g_A = [1, 1], g_B = [1, 0]$, and $g_C = [1, 0]$ at timestep $t^-$. We assume that the memory currently has zero value $[0, 0]$ for the node $C$ and some non-zero memory state for $A$ and $B$. At the end of time step $t-$, the memory of node $C$ gets updated to $s_C = [1, 0]$, and is reflected in the next timestep $t$. Now, suppose that an event occurs at timestep $t$ which changes the node feature of

$C$ to $g_C = [0,1]$. We note that $s_C$ is utilized to compute the updated $g_C^+$ or $g_B^+$, instead of the just current embeddings.

Now we analyze the message propagation using the standard energy transformer and compare it with our method. The updated node embeddings using our assumptions defined above are given below. For the energy transformer (**ET**)model:

$$g_{ET,B}^+ = g_{ET,B} + 2g_C \times D'_{B,C} \tag{14}$$

Whereas for our **TETN** case, the updated node embeddings are given as:

$$g_B^+ = g_B + 2(g_C + s_C) \times D_{B,C} \tag{15}$$

Here, $D_{B,C}$ and $D'_{B,C}$ are scalar values. $D'$ computes the similarity of node embeddings of nodes $B$ and $C$, whereas $D_{B,C}$ computes the similarity of memory infused node embeddings of $B$ and $C$, i.e, $g_B + s_B$ and $g_C + s_C$.

**Our Contribution** We would like to emphasize here that, when compared to standard energy transformers, which do not include memory due to their static nature, our energy-based formulation captures the memory of the neighbourhood when aggregating messages across hops. Our main contribution lies in the formulation of the energy function such that it allows message passing to include this memory while simultaneously minimizing the overall energy of the system via its monotonically decreasing nature.

Now we proceed to show how this additional memory term will affect the downstream task. Let us now check if there is a possibility of the existence of an edge between nodes $B$ and $C$. Ideally, we want the model to provide some weightage to the similarity of $g_B$ and $s_C$ at timestep $t^+$. This is because earlier, that is until $t-$, the nodes $B$ and node $C$ were similar, but at time $t$, an event updated the node features of node $C$, and now they are totally different (perpendicular, so similarity is 0). Ideally, we would like to give some weight to its previous state as well, which is captured by the memory module.

If we compute the similarity using the energy transformer (static) version, it clearly does not consider the contribution due to the high similarity of $g_B$ and $s_C$; it only computes similarity between the current node embeddings. On the other hand, when we compute the similarity using our method, one can clearly see all the contributions appear:

$$\begin{aligned}
\langle g_B^+, g_C^+ \rangle = & \langle g_B + 2(g_C + s_C)D_{B,C} \ , \ g_C + 2(g_B + s_B)D_{B,C} \rangle \\
= & \alpha_1 \langle g_B, g_C \rangle + \alpha_2 \langle s_B, s_C \rangle + \alpha_3 \langle s_B, g_C \rangle \\
& + \alpha_4 \langle g_B, s_C \rangle + \alpha_5 \langle g_B, s_B \rangle + \alpha_6 \langle g_C, s_C \rangle + c \,,
\end{aligned}$$

where $\alpha_i$ for $i \in \{1,2,3,4,5,6\}$ and $c$ are constants.

The above equation clearly shows that, using our energy-based formulation, one obtains a way to use the previous memory state of the nodes as well to effectively contribute to the node embeddings' updates, thereby ensuring long-range propagation in time. Moreover, the additional terms do not change the monotonicity of the energy formulation and hence the ODE updates ensure a navigation of the energy landscape towards lower energy configurations.

### A.3 Data Flow for a CTDG event

For more clarity on the **TETN** architecture, we show below the story of one CTDG event (e.g. a retweet from User A to User B) flowing through our network step by step, see Figure 6.

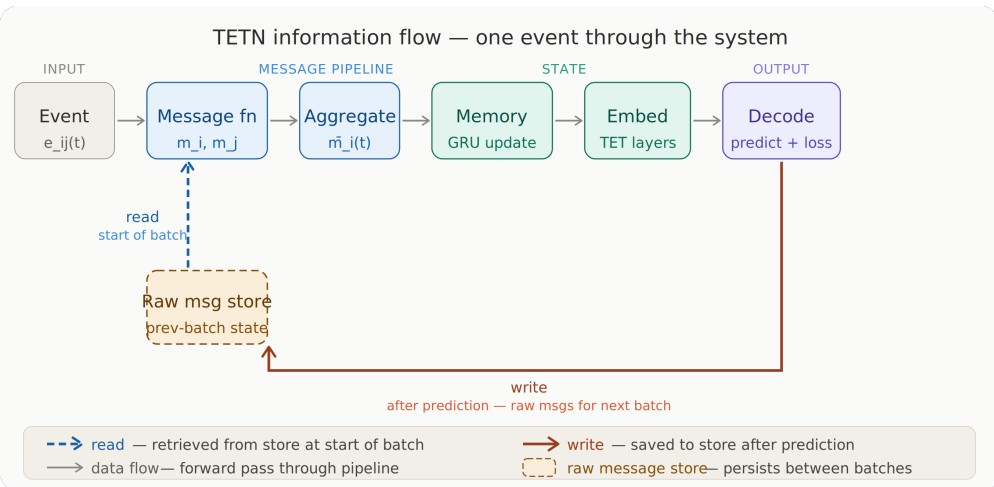

Figure 6: **TETN** flow of one CTDG event

**I - A retweet arrives:** User A retweets User B's post at time $t$. This generates a timestamped interaction event $e_{AB}(t)$ containing both node IDs, feature for the edge (as embedding), and a timestamp $t$. This is the raw signal that will flow through every **TETN** module.

**II - Retrieve past raw messages:** Before updating anything, **TETN** looks up the Raw Message Store for the last stored raw messages for nodes A and B, $rm_A(t^-)$ and $rm_B(t^-)$, respectively. These come from the previous event(s) — using past data (not the current event) avoids information leakage while still letting memory modules receive gradients during training.

**III - Compute messages:** The message function takes each node's raw messages and computes the message for updating the memory. It produces two messages — one for $A$ as source, $m_A(t)$, one for $B$ as destination, $m_B(t)$ — encoding what the past event meant for each node.

**IV - Aggregate messages:** If $A$ or $B$ appeared in multiple events within the same batch, their messages are aggregated before the memory update. This collapses multiple signals into a single $\overline{m}(t)$ per node, ready for the GRU.

**V - Update memory:** The *mem* takes the aggregated message and the old memory to produce updated memory vectors $s_A(t)$ and $s_B(t)$. This vector is **TETN**'s long-term summary of $A$ and $B$'s full interaction history. This module is what helps in information propagation over time, relative to memory-free variants.

**VI - Graph Propagation Module:** The current node features $g_A(t)$ and $g_B(t)$ and the updated memory states $s_A(t)$ and $s_B(t)$ are passed through the **TET** layers to generate final embeddings. With updated memories, multi-head energy attention aggregates information from $A$'s most recent temporal neighbours. $A$'s embedding (current and memory) is the query; each neighbour contributes key, their memory, edge features and time encoding $\phi(t)$. This produces an up-to-date embedding even if $A$ was recently inactive — solving the staleness problem.

**VII - Decode and compute loss:** A simple MLP decoder takes the final concatenated embeddings $x_A(t)$ and $x_B(t)$ and outputs the edge probability. BCE loss is computed against the true label; gradients flow back through the embedding and memory modules.

**VIII - Compute and store raw Messages** The raw message function takes each node's previous memory $s(t^-)$, the time gap $\Delta t$ since their last event, and the current edge features $e_{AB}(t)$. In practice it is the identity (concatenation). It produces two raw messages, $rm_A(t)$ and $rm_B(t)$ encoding what the current event means for each node. Then, update the raw message store with $rm_A(t)$ and $rm_B(t)$.

## A.4 AUC-ROC Results

We report the Area under the Receiver Operating Characteristic curve (AUC-ROC) values for the experiments on CTDG datasets in Tables 3 and 4. These results are consistent with the results in Tables 1 and 2.

Table 3: Results of the future link prediction task - transductive setting. We report the mean test set area under the ROC curve (AUC-ROC) and standard deviation in percent averaged over random weight initializations. Methods are ranked by average rank across all datasets (lower is better).

| Method ↓ / Dataset → | Wikipedia | UCI | Reddit | MOOC | Enron | LastFM | Avg. Rank |
|---|---|---|---|---|---|---|---|
| TETN (ours) | $\mathbf{99.15}_{\pm 0.13}$ | $\mathbf{98.95}_{\pm 0.10}$ | $\mathbf{99.32}_{\pm 0.02}$ | $84.70_{\pm 0.40}$ | $\mathbf{93.56}_{\pm 1.05}$ | $\mathbf{96.28}_{\pm 0.57}$ | **1.83** |
| DyGMamba | $99.08_{\pm 0.02}$ | $94.77_{\pm 0.18}$ | $99.27_{\pm 0.01}$ | $89.58_{\pm 0.12}$ | $93.34_{\pm 0.23}$ | $93.31_{\pm 0.18}$ | **2.00** |
| DyGFormer | $98.92_{\pm 0.03}$ | $94.45_{\pm 0.22}$ | $99.15_{\pm 0.01}$ | $88.08_{\pm 0.50}$ | $93.20_{\pm 0.12}$ | $93.03_{\pm 0.11}$ | **3.00** |
| TGN | $98.37_{\pm 0.10}$ | $92.03_{\pm 0.69}$ | $98.61_{\pm 0.05}$ | $\mathbf{91.91}_{\pm 0.82}$ | $88.72_{\pm 0.95}$ | $76.64_{\pm 4.66}$ | **3.67** |
| CTAN | $97.00_{\pm 0.21}$ | $76.25_{\pm 2.83}$ | $97.24_{\pm 0.75}$ | $85.40_{\pm 2.67}$ | $87.09_{\pm 1.51}$ | $85.12_{\pm 0.77}$ | 6.00 |
| TGAT | $96.60_{\pm 0.07}$ | $78.76_{\pm 1.10}$ | $98.50_{\pm 0.01}$ | $87.01_{\pm 0.16}$ | $68.57_{\pm 1.46}$ | $71.47_{\pm 0.14}$ | 6.50 |
| JODIE | $96.36_{\pm 0.14}$ | $90.35_{\pm 0.51}$ | $98.29_{\pm 0.05}$ | $84.50_{\pm 0.87}$ | $87.77_{\pm 2.43}$ | $70.89_{\pm 1.97}$ | 6.83 |
| GraphMixer | $96.89_{\pm 0.04}$ | $91.62_{\pm 0.52}$ | $97.17_{\pm 0.02}$ | $84.04_{\pm 0.12}$ | $84.16_{\pm 0.34}$ | $73.51_{\pm 0.14}$ | 7.00 |
| DyRep | $94.43_{\pm 0.32}$ | $69.46_{\pm 2.66}$ | $98.13_{\pm 0.04}$ | $84.50_{\pm 0.60}$ | $83.09_{\pm 2.20}$ | $71.40_{\pm 2.12}$ | 8.00 |

Table 4: Results of the future link prediction task - inductive setting. We report the mean test set area under the ROC curve (AUC-ROC) and standard deviation in percent averaged over random weight initializations. Methods are ranked by average rank across all datasets (lower is better).

| Method ↓ / Dataset → | Wikipedia | UCI | Reddit | MOOC | Enron | LastFM | Avg. Rank |
|---|---|---|---|---|---|---|---|
| DyGMamba | $98.72_{\pm 0.03}$ | $92.70_{\pm 0.19}$ | $98.88_{\pm 0.01}$ | $89.34_{\pm 0.12}$ | $\mathbf{89.76}_{\pm 0.21}$ | $94.36_{\pm 0.13}$ | **1.83** |
| TETN (ours) | $\mathbf{98.81}_{\pm 0.05}$ | $\mathbf{95.93}_{\pm 0.21}$ | $\mathbf{98.92}_{\pm 0.10}$ | $80.47_{\pm 0.50}$ | $89.61_{\pm 1.10}$ | $\mathbf{97.35}_{\pm 0.35}$ | **2.33** |
| DyGFormer | $98.49_{\pm 0.02}$ | $92.43_{\pm 0.20}$ | $98.70_{\pm 0.02}$ | $87.75_{\pm 0.42}$ | $89.59_{\pm 0.10}$ | $94.10_{\pm 0.09}$ | **3.00** |
| TGN | $97.71_{\pm 0.19}$ | $86.27_{\pm 1.49}$ | $97.30_{\pm 0.12}$ | $\mathbf{91.58}_{\pm 0.74}$ | $79.40_{\pm 1.77}$ | $82.61_{\pm 2.62}$ | 4.00 |
| JODIE | $94.43_{\pm 0.28}$ | $78.78_{\pm 1.11}$ | $96.42_{\pm 0.13}$ | $83.82_{\pm 0.30}$ | $80.16_{\pm 1.50}$ | $82.49_{\pm 0.94}$ | 5.67 |
| GraphMixer | $96.26_{\pm 0.04}$ | $89.26_{\pm 0.42}$ | $94.95_{\pm 0.08}$ | $82.76_{\pm 0.13}$ | $76.08_{\pm 0.92}$ | $80.34_{\pm 0.14}$ | 6.17 |
| TGAT | $95.93_{\pm 0.19}$ | $77.41_{\pm 0.65}$ | $97.02_{\pm 0.04}$ | $86.67_{\pm 0.24}$ | $64.25_{\pm 1.29}$ | $76.76_{\pm 0.22}$ | 6.33 |
| DyRep | $91.31_{\pm 0.40}$ | $58.84_{\pm 2.54}$ | $95.87_{\pm 0.21}$ | $83.42_{\pm 0.77}$ | $75.82_{\pm 3.14}$ | $82.82_{\pm 1.17}$ | 6.83 |
| CTAN | $92.59_{\pm 0.70}$ | $48.58_{\pm 6.02}$ | $82.35_{\pm 4.03}$ | $66.38_{\pm 1.59}$ | $61.49_{\pm 2.78}$ | $75.23_{\pm 2.24}$ | 8.83 |

## A.5 Statistical Significance Tests

We observe that for datasets Wikipedia and Reddit, the margin for improvement of AP/AUC-ROC scores is negligible relative to the top baselines. From Tables 1, 2, 3 and 4, we see that our model showcases better or overlapping mean metrics relative to the baseline DyGMamba, with higher standard deviation in some cases. To provide a better evidence of our results, we perform a statistical significance test to report the confidence interval for our model's mean AP. We perform the Welch's two-sample t-test (one-tailed), which assumes unequal population variances. Hypotheses (one-tailed, right-sided) for the test is:

$$H_0 : \mu_1 - \mu_2 \leq 0 \quad H_1 : \mu_1 - \mu_2 > 0$$

where $\mu_1$ is our model's true mean and $\mu_2$ is the top baseline's (DyGMamba) true mean. The sample means for the respective models from the Tables 1 - 4 are used for the test. The p-values are tabulated below in Table 5. We see better significance scores when we consider the AUC-ROC scores, since the standard deviations are lower. We see that the Hypothesis $H_0$ can be rejected with a significance level of $\alpha = 0.2$, when we consider the AUC-ROC scores. The observed difference is so large that it would occur with probability less than $\alpha = 0.2$ under $H_0$. That implies that our model performs better than the baseline, that is, $\mu_1 > \mu_2$,

with 80% probability. We note that with n=5, the t-distribution has only 4 degrees of freedom, making the test conservative and of low statistical power.

Table 5: p-values for the one-tailed Hypotheses with significance level $\alpha = 0.2$.

| Metric | Test Setting → Dataset ↓ | Transductive | Inductive |
|---|---|---|---|
| AP | Wikipedia | **0.0449** | 0.3036 |
| | Reddit | 0.3811 | 0.3642 |
| AUC-ROC | Wikipedia | **0.1485** | **0.0095** |
| | Reddit | **0.0013** | 0.2114 |

## A.6 TGBL Dataset

In Table 6, we present the MRR scores obtained for the link prediction task on tgbl-wiki and tgbl-review datasets discussed in Huang et al. (2023). We observe that on the large-scale tgbl-review dataset (with $4,873,540$ edges), **TETN** performs better than TGN and DyGFormer. This again emphasizes our model's ability to deal with large-scale, long-range datasets. This is in alignment with our results on CTDG datasets such as, LastFM.

Table 6: MRR values for tgbl-v2 datasets.

| Dataset → Method ↓ | tgbl-wiki | tgbl-review |
|---|---|---|
| DyGFormer | $\mathbf{0.79_{\pm 0.004}}$ | $0.224_{\pm 0.015}$ |
| TETN (ours) | $0.61_{\pm 0.02}$ | $\mathbf{0.375_{\pm 0.003}}$ |
| TGN | $0.39_{\pm 0.06}$ | $0.349_{\pm 0.020}$ |

## A.7 Ablation Studies

In this section, we present the ablation results for different parts of the model. Tables 7 and Table 8 show the impact of Hopfield layers and the use of edges in TET. We see that each component is essential for our model. Switching off any of the components leads to a drop in the average precision metric.

We also tried ablation experiments *without using the energy attention* component, i.e., by considering $E = E^{HN}$, without $E^{ATT}$. This led to a significant drop in precision and made the model behave as a random predictor on the current dataset.

Table 7: Average Precision scores for different model configurations showing the impact of Hopfield Network, and edge features.

| Dataset | No HN | No Edges | Full Model |
|---|---|---|---|
| UCI (trans.) | $94.05_{\pm 0.20}$ | $93.72_{\pm 0.01}$ | $\mathbf{98.98_{\pm 0.03}}$ |
| UCI (ind.) | $91.36_{\pm 0.60}$ | $80.82_{\pm 0.12}$ | $\mathbf{95.44_{\pm 0.15}}$ |
| Wikipedia (trans.) | $97.91_{\pm 0.30}$ | $92.15_{\pm 0.03}$ | $\mathbf{99.20_{\pm 0.01}}$ |
| Wikipedia (ind.) | $97.54_{\pm 0.30}$ | $91.43_{\pm 0.05}$ | $\mathbf{98.83_{\pm 0.13}}$ |

Table 8: ROC-AUC scores for different model configurations showing the impact of Hopfield Network, and edge features

| Dataset | No HN | No Edges | Full Model |
|---|---|---|---|
| UCI (trans.) | $94.94_{\pm 0.02}$ | $94.56_{\pm 0.01}$ | $\mathbf{98.95_{\pm 0.10}}$ |
| UCI (ind.) | $91.90_{\pm 0.07}$ | $84.96_{\pm 0.01}$ | $\mathbf{95.93_{\pm 0.21}}$ |
| Wikipedia (trans.) | $97.83_{\pm 0.03}$ | $93.58_{\pm 0.02}$ | $\mathbf{99.15_{\pm 0.05}}$ |
| Wikipedia (ind.) | $97.40_{\pm 0.03}$ | $92.67_{\pm 0.03}$ | $\mathbf{98.81_{\pm 0.01}}$ |

## A.8 Hyperparameter Studies

We evaluated our model by varying the hyperparameters $\gamma_1$ and $\gamma_2$ and report the results in Table 9. The experiment is conducted on tgbl-wiki v2 dataset. This highlights the effect of edge term (controlled by $\gamma_2$) in the ODE update equation (Eq. 11). We clearly see that the additional gradients of energy with respect to the memory states, $\frac{\partial E}{\partial s}$, and with respect to the edges, $\frac{\partial E}{\partial e}$, both play an essential role in the update equations.

Table 9: Ablation on $\gamma_1, \gamma_2$ on tgbl-wiki

| $\gamma_1$ | $\gamma_2$ | MRR |
|---|---|---|
| 0.0 | 0.0 | 0.31 |
| 0.0 | 0.5 | 0.61 |
| 0.5 | 0.0 | 0.39 |
| 0.5 | 0.5 | 0.58 |

## A.9 Complexity Analysis

We analyze the computational complexity of TET layer forward pass, where $D$ is the feature dimension, $Y$ is the hidden dimension of attention, $K$ is the number of neighbours over $k$-hop. Let $n, \mathbf{e}$ denotes the number of nodes, edges of the graph, and $H$ is the number of attention heads.

**Energy Attention (Eq. 9)**    For a single head, the energy attention involves: (1) query-key projections $\mathcal{O}(nDY)$, (2) $A_{BC}$ term involves $\mathcal{O}(nKY)$, and (3) aggregation of the energy attention term involves $\mathcal{O}(nD)$ computations.

**Hopfield network (Eq. 4)**    We use the $L$-layered MLP-based Hopfield network used in the implementation of energy transformer (Hoover et al., 2024) which has a complexity of $\mathcal{O}(nD^2L)$.

Overall run time complexity for forward pass: $\mathcal{O}(nDY) + O(nYK) + O(nD^2L)$.

Considering a single CTDG event and a single layer, the forward-pass complexity of the `TETN`, DyGFormer, and DyGMamba models are as below.

$$
\begin{aligned}
\texttt{TETN}: &\quad \mathcal{O}(DY + YK + D^2), \\
\text{DyGFormer}: &\quad \mathcal{O}(K^2 D + KD^2), \\
\text{DyGMamba}: &\quad \mathcal{O}(KD\,d_{\text{ssm}}).
\end{aligned}
$$

We note that $d_{\text{ssm}}$ of the state-space model DyGMamba is equivalent to the hidden dimension, $Y$ of the `TETN` model.

**Memory complexity for forward pass of `TETN`:** $\mathcal{O}(n + \mathbf{e})$, which is identical to that of Graph Attention Networks (Veličković et al., 2017).

Table 10: Comparison of peak memory utilization and training time per epoch across datasets for TET with DyGFormer (Yu et al., 2023), and DyGMamba (Ding et al., 2024).

| | Peak memory (GB) | | | Training time per epoch (min) | | |
|---|---|---|---|---|---|---|
| Method | UCI | Enron | LastFM | UCI | Enron | LastFM |
| DyGMamba | 1.93 | 2.74 | 4.17 | 0.60 | 2.05 | 28.45 |
| DyGFormer | 2.30 | 3.23 | 7.57 | 0.62 | 2.73 | 47.00 |
| TETN (ours) | 2.05 | 1.46 | 3.99 | 1.265 | 2.10 | 36.34 |

Empirical results on the run-time and memory usage of the TET graph embedding module during training are presented in Table 10. `TETN` achieves a favourable trade-off between memory efficiency and computation time. On the Enron and LastFM datasets, `TETN` exhibits notably lower peak memory consumption compared to DyGFormer (1.46 GB vs. 3.23 GB on Enron, and 3.99 GB vs. 7.57 GB on LastFM), highlighting its ability to handle large dynamic graphs more efficiently. While DyGMamba maintains slightly lower runtime on smaller datasets such as UCI, the difference diminishes as dataset size and temporal density increase, indicating that `TETN` scales competitively for real-world temporal graphs.

## A.10 Dataset Statistics

The statistics of the CTDG datasets used in our experiments are given below in Table 11. Each dataset is split chronologically, in the ratio of $70\% - 15\% - 15\%$, for training, validation, and testing, respectively.

Table 11: Statistics of the datasets used in our experiments

| | # Nodes | # Edges | # Edge ft. | # Node ft. | # Duration | # Time Stamps |
|---|---|---|---|---|---|---|
| Wikipedia | $9,227$ | $157,474$ | 172 | 0 | 1 month | $152,757$ |
| UCI | $1,899$ | $59,835$ | 172 | 0 | 196 days | $58,911$ |
| Reddit | $10,984$ | $672,447$ | 172 | 0 | 1 month | $669,065$ |
| Enron | 184 | $125,235$ | 0 | 0 | 3 years | $22,632$ |
| MOOC | $7,144$ | $411,749$ | 4 | 0 | 17 months | $345,600$ |
| LastFM | $1,980$ | $1,293,103$ | 0 | 0 | 1 month | $1,283,614$ |

## A.11 Hyper-parameter Space

The grid of hyper-parameters used in our experiments is listed in Table 12.

Table 12: Hyper-parameter Space

| Hyper-parameter | Values |
|---|---|
| # layers | {1, 2, 3} |
| learning rate | {1e-3, 1e-4} |
| dropout | {0.1} |
| Time encoding dim | {100} |
| Node embedding dim, $D$ | {172} |
| Edge embedding dim, $D$ | {172} |
| Hidden dim (in Attention), $Y$ | {172} |
| # Heads, $H$ | {2} |
| $\epsilon$ | { 0.5, 1.0} |
| $\gamma_1$, $\gamma_2$ | {1.0} |

### A.12 Algorithm for Training and Inference

---

**Algorithm 1** $\mathbf{emb}_t(x, \hat{s}, e, u, v, w)$

---

1: Layer parameters $W^K, W^Q \in \mathbf{R}^{Y \times D}$
2: $x^{(1)} \leftarrow x$
3: $\mathcal{V} \leftarrow \{u, v, w\}$ {$w$ represents the node from the negative edge sample}
4: **for all** $l = 1, 2 \ldots L$ **do**
5: $\quad$ TET Block Starts
6: $\quad \mathcal{V} \leftarrow \mathcal{V} \cup \eta_u^{(l)}(0,t) \cup \eta_v^{(l)}(0,t) \cup \eta_w^{(l)}(0,t)$ { $\eta_u^{(l)}(0,t)$ is the set of $l$-hop neighbours of node $u$ till time $t$.}
7: $\quad g^{(l)} \leftarrow \mathrm{LayerNorm}(x^{(l)})$
8: $\quad s \leftarrow \mathrm{LayerNorm}(\hat{s})$ {$s$ is a local variable}
9: $\quad \left. \begin{array}{l} K_{\alpha B} \leftarrow \sum_j W_{\alpha j}^K (g_{Bj}^{(l)} + s_{Bj}) \\ Q_{\alpha B} \leftarrow \sum_j W_{\alpha j}^Q (g_{Bj}^{(l)} + s_{Bj}) \end{array} \right\}$ $\left\{ \substack{\textbf{Memory infusion} \\ (\forall B \in \mathcal{V}, \alpha \in D)} \right\}$
10: $\quad E \leftarrow E^{ATT} + E^{HN}$ {**Energy attention** and **Hopfield Energy** are computed.}
11: $\quad \left. \begin{array}{l} x_u^{(l+1)} \leftarrow g_u^{(l)} - \eta \nabla_{g_u, s_u} E \\ x_v^{(l+1)} \leftarrow g_v^{(l)} - \eta \nabla_{g_v, s_v} E \\ x_w^{(l+1)} \leftarrow g_w^{(l)} - \eta \nabla_{g_w, s_w} E \end{array} \right\}$ $\left\{ \substack{\textbf{Token updation} \\ \text{using energy gradients}} \right\}$
12: $\quad$ TET Block Ends
13: **end for**
14: return $x_u^{(L)}, x_v^{(L)}, x_w^{(L)}$

---

**Algorithm 2** Link Prediction using TET

---

1: $\mathbf{s} \leftarrow \mathbf{0}$ {Initialize memory to zeros}
2: $\mathbf{rm} \leftarrow []$ {Initialize raw messages}
3: **for all** batch $(i, j, x, e, t) \in$ training data **do**
4: $\quad \mathbf{n} \leftarrow$ sample negatives
5: $\quad \mathbf{m} \leftarrow \mathrm{msg}(\mathbf{rm})$ {Compute messages from raw features}
6: $\quad \tilde{\mathbf{m}} \leftarrow \mathrm{agg}(\mathbf{m})$ {Aggregate messages for the same nodes}
7: $\quad \hat{\mathbf{s}} \leftarrow \mathrm{memory\_updater}(\tilde{\mathbf{m}}, \mathbf{s})$ {Get updated memory}
8: $\quad z_i, z_j, z_n \leftarrow emb_t(x, \hat{s}, e, i, j, n)$ {Compute node embeddings} (Algorithm 1).
9: $\quad \mathbf{p}_{pos}, \mathbf{p}_{neg} \leftarrow \mathrm{dec}(\mathbf{z}_i, \mathbf{z}_j), \mathrm{dec}(\mathbf{z}_i, \mathbf{z}_n)$ {Compute interactions probs}
10: $\quad l = \mathrm{BCE}(\mathbf{p}_{pos}, \mathbf{p}_{neg})$ {Compute BCE loss}
11: $\quad \mathbf{rm}_i, \mathbf{rm}_j \leftarrow (\hat{\mathbf{s}}_i, \hat{\mathbf{s}}_j, t, e), (\hat{\mathbf{s}}_j, \hat{\mathbf{s}}_i, t, e)$ {Compute raw messages}
12: $\quad \mathbf{rm} \leftarrow \mathrm{store\_raw\_messages}(\mathbf{rm}_i, \mathbf{rm}_j)$ {Store raw messages}
13: $\quad \mathbf{s}_i, \mathbf{s}_j \leftarrow \hat{\mathbf{s}}_i, \hat{\mathbf{s}}_j$ {Store updated memory for sources and destinations}
14: **end for**

---

### A.13 Baseline Models for Temporal and Static Graph Machine Learning

The majority of methods for deep learning on graphs assume that the underlying graph is static. However, most real-world interaction systems, such as social networks or biological interactions, are dynamic. Learning on dynamic graphs, as in works such as Sankar et al. (2019), is limited to the setting of discrete-time dynamic graphs represented as a sequence of graph snapshots. Such approaches are unsuitable for many practical real-world settings, such as social networks, where dynamic graphs are continuous (i.e., edges can appear at any time) and evolving (i.e., new nodes join the graph continuously). Several works have been proposed to address this setting, among them (Trivedi et al., 2018; Kumar et al., 2019; Xu et al., 2020; Rossi et al., 2020; Cong et al., 2023). Since the continuous-time dynamic graph setting is the more general one, we provide a high-level overview of select methods below to aid in a better understanding of our approach:

1. **Temporal Graph Networks (TGN)** (Rossi et al., 2020): TGN is a method for temporal graph machine learning on continuous-time dynamic graphs, where the graph is represented as a se-

quence of time-stamped events (e.g., edge additions, node updates). The framework consists of four core modules: (1) a *Memory Module* that maintains a compressed state vector $\mathbf{s}_i(t)$ per node, summarising its full interaction history; (2) a *Message Function* that computes update signals $\mathbf{m}_i(t) = \mathrm{msg}(\mathbf{s}_i(t^-), \mathbf{s}_j(t^-), \Delta t, \mathbf{e}_{ij}(t))$ for each event involving node $i$; (3) a *Message Aggregator* that consolidates multiple messages for the same node within a batch into a single signal $\bar{\mathbf{m}}_i(t)$; and (4) a *Graph Embedding Module* that generates temporal node representations by attending over a node's temporal neighbourhood, incorporating neighbour memories, edge features, and time encodings to mitigate the memory staleness problem. To prevent information leakage, memory updates are performed using a *Raw Message Store* populated from previous batches before predicting interactions in the current batch.

2. **Energy Transformers (ET)** (Hoover et al., 2024): This energy-based method for static graph machine learning operates on the principle of energy function minimization. The energy function described in the work consists of two components:

   (a) **Multi-Head Energy Attention block** ($E^{ATTN}$): This block facilitates the exchange of information between tokens, wherein each token generates a pair of keys and queries. The primary objective of the energy attention function is to evolve the token representations such that the keys of a node become aligned with the queries of its neighbouring nodes, which are subsequently used for downstream tasks such as temporal link prediction.

   (b) **Hopfield Network block** ($E^{HN}$): This neural network acts as a content-addressable memory that refines token representations by aligning them with a set of learned memory patterns of node representations. Each token interacts with these stored memories and is updated toward the most compatible patterns in a continuous manner. In the ET method, the Hopfield layer is formulated as a structured variant of the standard transformer feed-forward layer with shared weights; however, various alternatives exist, including the classical Hopfield network Hopfield (1984a) and the modern continuous Hopfield network Ramsauer et al. (2020). Consequently, this module encourages token representations that are consistent with learned node representations, assigning lower energy to well-aligned configurations.

### A.14  Energy equations for the static graph as described in Hoover et al. (2024)

The components of the energy $E$ for static ET are as follows:

$$E^{ATT} = -\frac{1}{\beta}\sum_{h=1}^{N}\sum_{C=1}^{N} log\left(\sum_{B\in\mathcal{N}_C} \exp\left(\beta\sum_{\alpha} K_{\alpha hB}Q_{\alpha hC}\right)\right), \tag{16}$$

where key and query tensors are as follows :

$$K_{\alpha hB} = \sum_{j} W_{\alpha hj}^{K} g_{jB}, \mathbf{K} \in \mathbf{R}^{Y\times H\times N},$$

$$Q_{\alpha hC} = \sum_{j} W_{\alpha hj}^{Q} g_{jC}, \mathbf{Q} \in \mathbf{R}^{Y\times H\times N},$$

with $W^K \in \mathbf{R}^{Y\times H\times D}$ and $W^Q \in \mathbf{R}^{Y\times H\times D}$ being learnable parameters, and

$$E^{HN} = -\sum_{B=1}^{N}\sum_{\mu=1}^{K} G\left(\sum_{j}\xi_{\mu j}g_{jB}\right), \xi \in \mathbf{R}^{K\times D}. \tag{17}$$

## A.15 Notations

Table 13: Notations

| Notation | Description |
|---|---|
| $E$ | Total Energy |
| $E^{ATT}$ | Energy Attention |
| $E^{HN}$ | Hopfield Energy |
| $K$ | Key Tensor |
| $Q$ | Query Tensor |
| $F$ | Edge Tensor |
| $W^K, W^Q, W^F$ | Weight Matrices |
| $A, B, C$ | Node Indices |
| $N$ | Number of Nodes |
| $D$ | Node/Edge embedding dim |
| $Y$ | Hidden dim (in Attention) |
| $H$ | Number of Heads |
| $\epsilon$ | Time step in ODE |
| $\gamma_1, \gamma_2$ | Token update parameters |
| $m$ | Margin |

