# OpenReview forum: "Temporal Energy Transformer for Long Range Propagation in Continuous Time Dynamic Graphs"
_TMLR — Accepted by TMLR_

### Review · Reviewer_gdvq · 2026-03-06

**Summary Of Contributions:**

This paper introduces Temporal Energy Transformer Network (TETN), the first energy‑based architecture designed specifically for continuous‑time dynamic graphs (CTDGs). The authors argue that existing temporal GNN, especially attention‑based one, struggle with long‑range temporal dependencies and lack a principled theoretical foundation. To address this, they integrate ideas from energy-based models, modern Hopfield networks, and the Temporal Graph Network (TGN) framework.
The core idea is to define a learnable energy functional that depends on: the current node embedding $x_u(t)$, the node’s historical memory $S_u(t)$, and the representations of edges in its temporal neighborhood.
The model updates node embeddings by descending the energy landscape using an ODE‑based update rule. This ensures that the system evolves toward lower‑energy configurations, which the authors interpret as more temporally consistent representations. The architecture includes:
- Energy Attention (a modified attention mechanism infused with memory and edge features).
- Hopfield Energy (to retain long‑range information).
- Memory infusion (to integrate historical context into keys/queries) and a continuous‑time ODE update for token embeddings.

Empirically, TETN achieves state‑of‑the‑art performance on six widely used CTDG datasets (Wikipedia, UCI, Reddit, MOOC, Enron, LastFM), outperforming DyGFormer, DyGMamba, TGN, and others in both transductive and inductive link prediction. The authors emphasize that the energy formulation provides interpretability and theoretical grounding, and they show that the energy decreases monotonically over time.

**Audience:**

Yes

**Audience Explanation:**

The paper studies Continuous‑time dynamic graphs (CTDGs), which is an important research problem because they capture how the real world actually evolves. Most real systems don’t update in neat, fixed interval, they change whenever something happens. CTDGs are the mathematical and computational way to model that reality.

**Claims And Evidence:**

Yes

**Claims Explanation:**

-  Novel and principled energy-based formulation for temporal graphs. The paper is the first to define an explicit energy functional for CTDGs and integrate it into a temporal GNN. This is a meaningful conceptual contribution, not just an architectural tweak.
- Strong integration of TGN, energy transformers, and Hopfield networks. The architecture is thoughtfully designed: memory, attention, Hopfield energy, and ODE updates all interact coherently. The memory‑infused keys/queries and edge‑aware energy attention are particularly well‑motivated.
- Strong empirical results.

**Requested Changes:**

- The narrative oversells the theoretical contribution. The writing repeatedly claims “strong theoretical foundations, but the actual theoretical result is modest: a monotonic energy‑decrease guarantee under a specific ODE update.
- The paper lacks a clear, intuitive explanation of the energy functional. Add a simple, intuitive explanation before the formal definition. Provide a small toy example showing how energy changes, e.g., a simple 3‑node temporal graph.
- The architecture description is overly dense and hard to follow. Sections 3–5 read like a sequence of equations without a guiding narrative. The writing assumes the reader already understands TGN, ET, Hopfield networks, and CTDGs simultaneously. Add a high‑level overview figure showing the flow of information. Provide a story of one event flowing through the system.

---

### Review · Reviewer_Q33R · 2026-03-10

**Summary Of Contributions:**

The authors focus on representation learning for temporal graphs under dynamic phenomena, trying to enhance the ability of self-attention mechanisms in capturing long-range dependencies.
They specifically leverage energy-based models and propose a novel model, the so-called Temporal Energy Transformer (TET). Their approach relies on the Temporal Graph Network (TGN) framework and incorporates an energy-based graph propagation module, using a tailored energy functional to extract spatio-temporal information, coupled with a continuous-time differential equation for irregular dynamics. The employed TET layers use a dense associative memory model (similar to a Hopfield network). The model's efficacy is evaluated on several temporal graph datasets, showing competitive or superior performance in transductive and inductive dynamic link prediction scenarios, compared to recent SOTA models.

**Audience:**

Yes

**Audience Explanation:**

DL for temporal dynamic graphs constitutes a challenging domain in terms of representational capabilities, scalability, and interpretability and therefore, the proposed method constitutes a potentially interesting modern framework.

**Broader Impact Concerns:**

No ethical concerns.

**Claims And Evidence:**

No

**Claims Explanation:**

The authors make several claims concerning the novelty of their proposed method’s architectural design, their improved representational capacity, and experimental performance. The following main strengths of this work are substantially justified, as follows:
- **S1:** The evaluation for the task of temporal dynamic link prediction is thorough, incorporating comparisons on several datasets and baselines. The proposed TETN model obtained the best rank in terms of mean performance in the transductive and second-best in the inductive settings against 8 baselines and 7 datasets, showing performance improvements in several cases.
- **S2:** The experimental evaluation is accompanied by extensive ablation studies in the supplementary material, showing each component's contribution to the link prediction accuracy. Hyperparameter values are also studied.
- **S3:** A crucial complexity analysis is also included in the supplementary.
- **S4:** The proposed algorithm is nicely explained in terms of the main text and the supplementary, accompanied by analytical algorithms for the embedding and link prediction steps during training and inference.

However, the following weaker aspects can be identified with respect to the presentation of the proposed method and overall contribution:
- **W1:** The performance improvements of the proposed TETN method are, in several cases, incremental (see Reddit dataset in the transductive setting, Wikipedia in the inductive setting). There is no statistical significance test accompanying the results for the cases where there is overlap in performance due to high standard deviations of the proposed TETN method.
- **W2:** The authors claim (with some nice related works’ justification) that existing attention-based models (e.g., DyGFormer) struggle to capture long-term historical information in dynamic graphs; however, this is not clearly substantiated mathematically or empirically. There are several cases where DyGFormer is competitive or outperforms the proposed TETN (see Enron and LastFM in the inductive setting, Reddit and MOOC in the transductive setting). The claim about the improvement of capturing long-term dependencies with TET layers compared to vanilla self-attention is not adequately demonstrated.
- **W3:** The proposed method does not show any significant improvement in terms of computational complexity (memory and training time cost) compared to DyGMamba and DyGFormer, which underscores the possibility of the method scaling for very large graphs, particularly based on the imposed ODE updates, which are considered in the framework.
- **W4:** There is an overstated claim for the interpretability aspect of the proposed TETN method; however, it is not adequately justified via experiments/visualizations. It is interesting the statement of the authors that low-energy configurations can be interpreted as states where nodes are temporally and structurally consistent with their neighborhoods, but this should be backed up by practical examples. In the current form of the paper, none of the considered baselines and the proposed method are accompanied by any interpretability aspect with respect to the obtained representations.
- **W5:** There is a large body of literature on energy-based methods and Hopfield networks for time series and dynamical systems that could be used to strengthen the related works section. In the current form, for the non-expert audience in the energy-based models literature, it is hard to identify any novelty aspect of the energy attention in the proposed TET layers, which seem to heavily rely on the Energy Transformer (Hoover et al., 2024). The whole 3rd paragraph of page 11 makes statements that are not justified, e.g., “…adapting energy functionals to temporal graphs and addressing long-range propagation challenges remain non-trivial”.

**Requested Changes:**

Based on the previous comments, I suggest that the authors tackle the following aspects:

**[Based on W1 - Statistical Significance Tests]** Considering statistical significance tests to justify their claims about performance superiority, particularly for cases of overlapping performances with baselines and high stds. Justifying also the high stds among runs for specific cases is important.

**[Based on W2 - Example on Effectively Capturing Long-Range Dependencies]** A practical example showing where propagation in historical information through self-attention, compared to energy-based attention with the proposed TET layers, could justify the need for research beyond vanilla attention in temporal dynamic graphs.

**[Based on W4 - Example on Claimed Interpretability of Representations]** The paper makes strong claims regarding the interpretability of the representations produced by the proposed TETN model, not supported by empirical evidence. Could the authors provide a concrete example, visualization, or case study demonstrating how the learned representations can be interpreted in practice? Alternatively, if such evidence cannot be provided, the interpretability claims should be moderated.

**[Based on W5 - Energy-based Literature]** I suggest revisiting and expanding the related work on energy-based models and Hopfield networks. In its current form, the novelty of the proposed TET layers relative to prior work (e.g., Energy Transformer) is not clearly articulated. The authors should clarify what is new in their formulation and better justify some claims in the related work section with appropriate references.

---

### Review · Reviewer_CUWh · 2026-04-28

**Summary Of Contributions:**

This paper is well-written and easy to follow. The authors study long-range propagation in continuous-time dynamic graphs. They propose a temporal energy transformer.

**Audience:**

Yes

**Audience Explanation:**

The dynamic graph community may be interested in this paper.

**Claims And Evidence:**

Yes

**Claims Explanation:**

Sufficient experiments show the effectiveness of the proposals.

**Requested Changes:**

1. As the graph may be changed over time, it would be better to include more experiments to test if the proposed method is working in a streaming setting.
2. It is required to compare the efficiency, i.e., training time, inference time, and FLOPs, across the proposed method and baselines.
3. A case study is required to intuitively show the effectiveness of the proposals.

---

### Decision · Action_Editor_Rfam · 2026-06-12

**Recommendation:** Accept as is

**Audience:**

Yes

**Audience Explanation:**

The work addresses continuous-time dynamic graph learning, which is relevant to researchers studying temporal graphs and dynamic representation learning, which are among the TMLR audience.

**Claims And Evidence:**

Yes

**Claims Explanation:**

All reviewers agree that after revisions and authors' responses, the paper provides sufficient and convincing evidence.

In particular, the supporting evidence includes experiments on standard CTDG link prediction benchmarks in both transductive and inductive settings, comparisons to recent baselines such as TGN, DyGFormer, and DyGMamba, ablation studies, complexity analysis, and additional significance tests added during revision. Overall, these experiments support the claims made in the paper regarding the benefit given by energy-based propagation mechanism for continuous-time dynamic graphs,  and in particular improving long-range temporal representation learning within TGN frameworks.